# Risk of a seizure recurrence after a breakthrough seizure and the implications for driving: further analysis of the standard versus new antiepileptic drugs (SANAD) randomised controlled trial

L J Bonnett,[1] G A Powell,[2] C Tudur Smith,[1] AG Marson[2]

[1]Department of Biostatistics, University of Liverpool, Liverpool, UK
[2]Department of Molecular and Clinical Pharmacology, Clinical Sciences Centre, Liverpool, UK

**Correspondence to**
L J Bonnett; L.J.Bonnett@liverpool.ac.uk

## ABSTRACT

**Objectives** A breakthrough seizure is one occurring after at least 12 months seizure freedom while on treatment. The Driver and Vehicle Licensing Agency (DVLA) allows an individual to return to driving once they have been seizure free for 12 months following a breakthrough seizure. This is based on the assumption that the risk of a further seizure in the next 12 months has dropped <20%. This analysis considers whether the prescribed 1 year off driving following a breakthrough seizure is sufficient for this and stratifies risk according to clinical characteristics.

**Design, setting, participants, interventions and main outcome measures** The multicentre UK-based Standard versus New Antiepileptic Drugs (SANAD) study was a randomised controlled trial assessing standard and new antiepileptic drugs for patients with newly diagnosed epilepsy. For participants aged at least 16 with a breakthrough seizure, data have been analysed to estimate the annual seizure recurrence risk following a period of 6, 9 and 12 months seizure freedom. Regression modelling was used to investigate how antiepileptic drug treatment and a number of clinical factors influence the risk of seizure recurrence.

**Results** At 12 months following a breakthrough seizure, the overall unadjusted risk of a recurrence over the next 12 months is lower than 20%, risk 17% (95% CI 15% to 19%). However, some patient subgroups have been identified which have an annual recurrence risk significantly greater than 20% after an initial 12-month seizure-free period following a breakthrough seizure.

**Conclusions** This reanalysis of SANAD provides estimates of seizure recurrence risks following a breakthrough seizure that will inform policy and guidance about regaining an ordinary driving licence. Further guidance is needed as to how such data should be used.

**Trial registration number** SANAD is registered with the International Standard Randomised Controlled Trial Number Register ISRCTN38354748.

### Strengths and limitations of this study

▶ This reanalysis of Standard versus New Antiepileptic Drugs (SANAD) provides estimates of seizure recurrence risks following a breakthrough seizure that will inform policy and guidance about regaining an ordinary driving licence.
▶ The SANAD data largely reflect patients with newly diagnosed epilepsy so we have been unable to explore longer-term patterns of seizures.
▶ Patients with epilepsy may elect not to report breakthrough seizures to their clinicians or the relevant driving authority which may lead to an underestimation of risk.

## INTRODUCTION

A breakthrough seizure is defined as the first seizure after a minimum of 12 months seizure freedom while on treatment. The legislation[1] that directs the decisions of the UK Driver and Vehicle Licensing Agency (DVLA) is informed by a risk-based approach. This is summarised in guidance available on their website.[2] In The Motor Vehicle Regulation, epilepsy is defined as a history of two or more clinically unprovoked seizures.[1] According to this, people who have had a breakthrough seizure are usually allowed to regain their group 1 (ordinary) driving licence 1 year after the breakthrough seizure provided they have been seizure free, based on the assumption that their risk of a seizure in the next 12 months has fallen <20%. This minimum level of risk is supported by other European Union member states[3] and has been adopted in the criteria determining minimum driving standards that are being harmonised across the European Union. In the USA, each individual state has its own legislation for driving

with epilepsy and seizures. When surveyed in 2001,[4] most states (n=28) required people with epilepsy to have a time off driving (median 6 months, range 3–12 months), whereas in 19 states the time was decided by the treating doctor or a medical advisory board.

There are currently few published studies in which seizure recurrence risks are estimated and factors that modify risk investigated. Existing publications[5–8] have focused on recurrence immediately following a first seizure or recurrence after treatment withdrawal. Only Bonnett[7 8] has presented risks of recurrence in the next 12 months following seizure freedom at time points such as 6 or 12 months. At 6 months following a first seizure, the risk of another seizure in the next 12 months was 14% (10%–18%) for those who start antiepileptic drug treatment, and 18% (13%–23%) for those who do not.[7] At 3 months after withdrawal of antiepileptic drug treatment following at least 12 months remission from seizures, the risk of a seizure was 15% (10%–19%).[8] There are no publications considering risk of recurrence following breakthrough seizures. There is therefore a need for reliable published data to inform decisions made by clinicians, DVLA guidance and/or European Union legislation, and legislation outside the European Union.

The Standard versus New Antiepileptic Drugs (SANAD) trial compared standard and new antiepileptic drugs as monotherapy. Arm A recruited 1721 patients who were randomised to treatment with carbamazepine, gabapentin, lamotrigine, topiramate or oxcarbazepine. Arm B recruited 716 patients who were randomised to lamotrigine, topiramate or valproate. Patients were followed up to the end of the study whether they remained on their randomised treatment or not, according to the intention-to-treat principle. Outcomes assessed included time to 12-month remission, time to treatment failure and time to first seizure.

Here, data from a subset of participants achieving 12-month remission while on treatment followed by a breakthrough seizure have been analysed to estimate the subsequent risk of seizure recurrence. Modelling has been used to investigate how a number of clinical factors influence the outcome.

## METHODS
### Patients
The methods for the SANAD study have been published elsewhere.[9 10] In summary, patients were eligible for inclusion into SANAD if, in the previous year, they had a history of at least two clinically definite unprovoked epileptic seizures and they were at least 5 years of age. Patients were recruited into arm A if the recruiting clinician considered carbamazepine to be the optimal standard treatment option. Between 1 December 1999 and 1 June 2001, patients were allocated in a ratio of 1:1:1:1 to carbamazepine, gabapentin, lamotrigine and topiramate. From 1 June 2001 to 31 August 2004, an oxcarbazepine group was added to the trial and patients were randomly allocated in

a ratio of 1:1:1:1:1 to carbamazepine, gabapentin, lamotrigine, oxcarbazepine or topiramate.

Patients were eligible for inclusion in arm B if the recruiting clinician regarded valproate the standard treatment option. Participants were randomly allocated in a 1:1:1 ratio to valproate, lamotrigine or topiramate between 12 January 1999 and 31 August 2004. The two primary outcomes in SANAD were time to treatment failure from randomisation and time to the first period of 12 months of remission from seizures following randomisation.

In this paper, the arm A and arm B data sets have been combined in order to undertake prognostic modelling stratifying by arm. In the original publications trial arms were analysed and reported separately as the primary purpose was to compare the effectiveness of new antiepileptic drugs with the standard treatments. Here the purpose is different, the aim being to assess the risk of a seizure recurrence following a breakthrough seizure, irrespective of the specific drug that the patient was on at randomisation, or the subsequent choice of treatment.

In order to make the analysis reported here relevant to those of driving age, only participants who achieved 12-month remission while on treatment and then had a breakthrough seizure, and were aged ≥16 years when the breakthrough seizures occurred were included. Sixteen years of age was chosen as the lower cut-off as by the age 17, after 12 months of follow-up, they would be eligible for a provisional group 1 licence in the UK. Other European Union countries have a minimum driving age of 18 years[11] with some exceptions such as Hungary[12] and Southern Ireland,[13] where the limit is 17 years. In addition, the analysis only included patients who, in the 6 months prior to their breakthrough seizure, underwent an increase in dosage or had no change in dosage. In other words, patients with any decrease in dose either with an intention to withdraw, or not, were excluded as their seizure was likely to be due to antiepileptic drug withdrawal, which is handled differently in the legislation, and analyses informing legislation following antiepileptic drug withdrawal have been published.[8]

### Statistical analysis
The outcome of interest is the probability of a seizure recurrence in the next 12 months given that the participants have been seizure free from the breakthrough seizure to the time point in question. For example, the probability of someone who was seizure free for 6 months after his or her breakthrough seizure, having a seizure in months 7–18 was calculated by dividing the probability of having a seizure by 18 months by the probability of having a seizure by 6 months. Risks of recurrence in the next 12 months for other time points were calculated similarly using the Cox model. CIs for estimates were calculated using a revised version of Greenwood's formula.[14–16] Although SANAD was a randomised trial, in this analysis the outcome was measured from the date of the breakthrough seizure, not the date of randomisation.

Variables associated with a higher risk of seizure recurrence were determined univariably and after adjusting for multiple variables using log-rank tests and Cox proportional hazards modelling methods. A best-fitting, parsimonious, multivariable model was produced with variable reduction by Akaike's information criterion.[17] The recurrence risk in the next 12 months for combinations of risk factors was calculated from the multivariable model.[18] All analyses were undertaken using R V.3.2.3.

Continuous variables were investigated using log and fractional polynomial transformations.[19–22] The results for the continuous variables are presented as post hoc defined categorical variables with categories chosen according to knot positions for a spline model fit to the data.[23] Schoenfeld residual plots[24] and incorporation of time-dependent covariate effects were used to investigate the proportional hazards assumption. The predictive accuracy of the models was assessed using the c-statistic.[25]

Our list of potential prognostic factors included gender, febrile seizure history, first-degree relative with epilepsy, neurological insult, seizure type, epilepsy type, EEG result, CT or MRI result, total number of tonic–clonic seizures recorded prior to breakthrough seizure, age at breakthrough seizure, number of treatments required to achieve 12-month remission prior to breakthrough seizure (either monotherapy or polytherapy), time to achieve 12-month remission prior to breakthrough

seizure and breakthrough seizure treatment decision (no change to treatment plan, increase dosage or decrease dosage for any reason). The breakthrough seizure treatment decision is defined to have occurred up to 3 months after the seizure and is used as a proxy for the decision that was made at the time of first clinic visit following the breakthrough seizure.

Patients were classified as having neurological insult if they had learning disabilities or neurological deficit, while EEG was classified as normal, not clinically indicated, non-specific abnormality or epileptiform abnormality (focal or generalised spikes or spike and slow wave activity). Seizure types were classified according to the International League Against Epilepsy seizure classification.[26] Epilepsy type was first classified as focal, generalised or unclassified with the unclassified category being used when there was uncertainty between focal onset and generalised onset seizures.

## RESULTS

Figure 1 illustrates patient disposition of the 2627 patients recruited into both arms A and B of SANAD, and identifies patients relevant to this analysis; for the purposes of this analysis, data from both trial arms have been combined. Table 1 summarises the patient demographics for the 399 patients under analysis. Of these patients,

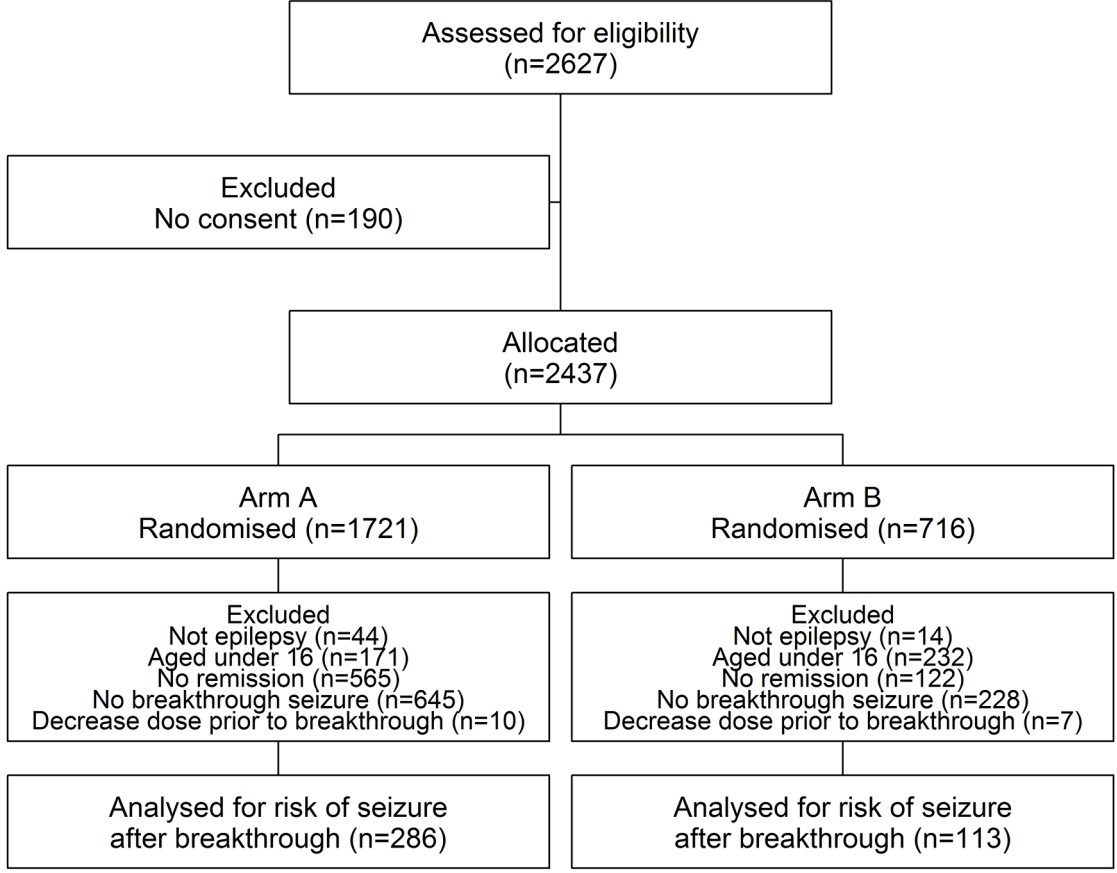

**Figure 1** Standard versus New Antiepileptic Drugs trial profile.

**Table 1**  Patient demographics

| Characteristics (n (%) unless otherwise stated) | Arm A (n=286) | Arm B (n=113) | Total (n=399) |
|---|---|---|---|
| Male | 159 (56) | 72 (64) | 231 (58) |
| Febrile seizure history | 15 (5) | 5 (4) | 20 (5) |
| Epilepsy in first-degree relative | 24 (8) | 21 (19) | 45 (11) |
| Neurological insult | 38 (13) | 9 (8) | 47 (12) |
| Seizures | | | |
| Simple or complex partial with secondary generalised seizures | 180 (63) | 5 (4) | 185 (46) |
| Simple or complex partial only | 72 (25) | 1 (0) | 73 (18) |
| Generalised tonic–clonic seizures only | 4 (1) | 32 (29) | 36 (9) |
| Absence seizures | 1 (0) | 5 (5) | 6 (2) |
| Myoclonic or absence seizures with tonic–clonic seizures | 0 (0) | 28 (25) | 28 (7) |
| Tonic–clonic seizures, uncertain if focal or generalised | 27 (10) | 34 (30) | 61 (15) |
| Other | 2 (1) | 8 (7) | 10 (3) |
| Epilepsy type | | | |
| Partial | 253 (88) | 6 (5) | 259 (65) |
| Generalised | 6 (2) | 69 (61) | 75 (19) |
| Unclassified | 27 (10) | 38 (34) | 65 (16) |
| EEG results | | | |
| Normal | 134 (47) | 32 (28) | 166 (42) |
| Non-specific abnormality | 49 (17) | 13 (12) | 62 (16) |
| Epileptiform abnormality | 69 (24) | 64 (57) | 133 (33) |
| Not clinically indicated | 34 (12) | 4 (3) | 38 (9) |
| CT/MRI scan results | | | |
| Normal | 164 (57) | 59 (52) | 223 (56) |
| Abnormal | 75 (26) | 10 (9) | 85 (21) |
| Not clinically indicated | 47 (17) | 44 (39) | 91 (23) |
| Number of treatments required to achieve 12-month remission | | | |
| Monotherapy | 219 (77) | 86 (77) | 305 (77) |
| Polytherapy | 67 (23) | 27 (23) | 94 (23) |
| Number of tonic–clonic seizures reported by first breakthrough seizure, median (IQR) | 3 (16) | 3 (26) | 3 (1–6) |
| Age at first breakthrough seizure, median (IQR) | 44.5 (31.857.7) | 24.0 (21.134.5) | 38.3 (24.5–53.5) |
| Time to achieve 12-month remission prior to breakthrough seizure (years), median (IQR) | 1.2 (1.01.9) | 1.1 (1.01.8) | 1.2 (1.0–1.9) |
| Treatment decision prior to breakthrough seizure | | | |
| No change to treatment plan | 261 (91) | 101 (89) | 362 (91) |
| Increase dosage | 25 (9) | 12 (11) | 37 (9) |
| Breakthrough seizure treatment decision | | | |
| No change to treatment plan | 169 (61) | 67 (61) | 236 (61) |
| Increase dosage | 99 (36) | 40 (37) | 139 (36) |
| Decrease dosage for any reason or missing decision | 9 (3) | 2 (2) | 11 (3) |

254 experienced at least one further seizure after breakthrough. Patients in arm A were followed up for a median of 1.67 years following a breakthrough seizure (IQR 0.85–2.59 years) while patients in arm B were followed up for a median of 1.41 years (IQR 0.55–2.56 years). In total, there were 705.6 patient-years of follow-up after the breakthrough seizure.

Figure 2 illustrates the risk of seizure recurrence after a breakthrough seizure. The median time to a further seizure following a breakthrough was 76 days (IQR

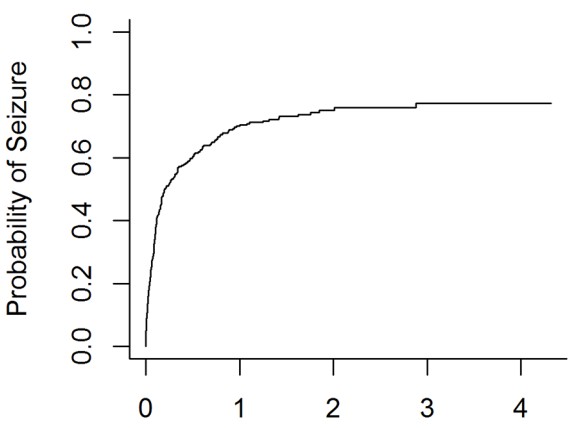

**Figure 2** Kaplan-Meier curve for time to next seizure following a breakthrough seizure.

57–122 days). The probability of a seizure by 12 months was 70.1%. In particular, 111 (28%) people had had a seizure by 1 month, 166 (42%) by 2 months, 214 (54%) by 6 months, 242 (61%) by 1 year, 252 (63%) by 2 years and 254 (64%) by the end of the follow-up period. Table 2 shows unadjusted 12-month seizure recurrence risks at various time points after the breakthrough seizure. At 6 months, the estimate is significantly >20%. At 12 months, however, the estimate is <20% and significantly so as the 95% CI does not include 20%.

Results for univariable and multivariable modelling of time to seizure recurrence are presented in table 3. In the univariable model, number of drugs required to achieve initial 12-month remission and time to achieve a first 12-month remission prior to breakthrough seizure were associated with seizure recurrence risk—patients requiring polytherapy to achieve remission were more likely to have a recurrence than those requiring mono-therapy. Additionally, patients achieving remission immediately at 1 year were less likely to have a recurrence following a breakthrough seizure than those who took longer to achieve 12-month remission. Breakthrough seizure treatment decision was also associated with the outcome; patients having an increase in dose after their breakthrough seizure were more likely to have a recurrence than those not changing their treatment, which

may be counterintuitive, but indicates clinicians are able to identify those at higher recurrence risk.

The final multivariable model included number of drugs required to achieve initial remission, time to achieve initial 12-month remission and breakthrough seizure treatment decision. As before, patients requiring polytherapy to achieve remission were more likely to have a recurrence than those requiring monotherapy, patients achieving remission immediately at 1 year were less likely to have a recurrence than those who took longer to achieve 12-month remission and patients increasing their dose after their breakthrough seizure were more likely to have a recurrence than those not changing their dose. There was no evidence to suggest that the proportional hazards assumption, underlying the Cox model, was invalid. The c-statistic for the model was 0.62, indicating that the model accurately discriminates participants 62% of the time, which is reasonable internal validation.[27 28]

Breakthrough seizure treatment decision, although significantly associated with the outcome, should not be considered as a modifiable variable as clinicians will find it very difficult to use this information to inform treatment decisions for future patients. Therefore, the model was refitted excluding this covariate, and the resulting parsimonious model included number of drugs attempted to achieve initial 12-month remission and time taken to achieve initial 12-month remission. The direction of the effects remained unchanged (table 3).

The risk of recurrence at 12 months for patients with particular characteristics was estimated from the parsimonious multivariable regression model. Results can be seen in table 4. At 6 months seizure freedom following a breakthrough seizure, no patient subgroup had a risk of recurrence that was <20%. By 12 months of seizure freedom, the current recommended time off driving following a breakthrough seizure, several patient subgroups still had estimates in excess of the 20%. In particular, the length of time required for the estimate of seizure recurrence to fall <20% for patients requiring polytherapy to achieve initial 12-month remission, and taking ≥3 years to enter initial period of 12-month remission is 15 months.

## DISCUSSION

In the UK, the DVLA prescribes 1 year off driving following a breakthrough seizure based on legislation and the assumption that a person's risk of a seizure in the next 12 months is <20%. According to data from the SANAD study, the overall risk of a seizure recurrence, unadjusted for any covariates, falls significantly <20% by 12 months of seizure freedom following the breakthrough seizure as required. Covariates significantly associated with the outcome were time taken to achieve an initial 12-month remission, number of drugs required to achieve that remission and breakthrough seizure treatment decision. As expected, those patients who achieve a period of 12-month remission

**Table 2** Unadjusted 12-month seizure recurrence risks at time points after breakthrough seizure: risk (%, 95% CI)

| Time seizure free after breakthrough seizure (months) | Number at risk | Risk of seizure in following 12 months. % |
|---|---|---|
| 6 | 119 | 32 (28 to 36) |
| 9 | 99 | 24 (21 to 27) |
| 12 | 80 | 17 (15 to 19) |

**Table 3** Effect estimates from univariable and multivariable models

| Variable | Comparison | Univariable p value | Univariable HR (95% CI) | Multivariable HR (95% CI) | Multivariable HR (95% CI) w/o decision variable |
|---|---|---|---|---|---|
| Gender | Female | 0.43 | 1.00 | N/A | N/A |
| | Male | | 1.11 (0.86 to 1.42) | | |
| Febrile seizure history | Absent | 0.28 | 1.00 | N/A | N/A |
| | Present | | 0.69 (0.35 to 1.34) | | |
| Epilepsy in first-degree relative | Absent | 0.82 | 1.00 | N/A | N/A |
| | Present | | 1.05 (0.69 to 1.59) | | |
| Neurological insult | Absent | 0.59 | 1.00 | N/A | N/A |
| | Present | | 0.90 (0.62 to 1.32) | | |
| Seizure type | Simple/complex partial+secondary gen. | | 1.00 | N/A | N/A |
| | Simple/complex partial only | 0.35 | 1.17 (0.84 to 1.63) | | |
| | Generalised TC only | 0.65 | 0.87 (0.48 to 1.58) | | |
| | Absence | 0.96 | 1.03 (0.35 to 3.03) | | |
| | Myoclonic/absence+TC | 1.00 | 1.00 (0.49 to 2.03) | | |
| | TC (uncertain if focal or gen.) | 0.49 | 0.85 (0.54 to 1.34) | | |
| | Other | 0.89 | 1.07 (0.44 to 2.55) | | |
| Epilepsy type | Partial | | 1.00 | N/A | N/A |
| | Generalised | 0.65 | 0.88 (0.52 to 1.49) | | |
| | Unclassified | 0.55 | 0.88 (0.57 to 1.35) | | |
| EEG results | Normal | | 1.00 | N/A | N/A |
| | Non-specific abnormality | 0.62 | 0.91 (0.63 to 1.32) | | |
| | Epileptiform abnormality | 0.87 | 0.98 (0.73 to 1.30) | | |
| | Not done/missing | 0.05 | 0.60 (0.36 to 1.00) | | |
| CT/MRI scan results | Normal | | 1.00 | N/A | N/A |
| | Abnormal | 0.15 | 0.79 (0.57 to 1.09) | | |
| | Not done/missing | 0.86 | 0.97 (0.71 to 1.33) | | |
| Number of drugs attempted for remission | Monotherapy | 0.01 | 1.00 | 1.00 | 1.00 |
| | Polytherapy | | 1.47 (1.11 to 1.94) | 1.37 (1.02 to 1.84) | 1.28 (0.96 to 1.71) |
| Number of TC seizures reported by first breakthrough seizure (linear) | 0 | 0.60 | 1.00 | N/A | N/A |
| | 1 | | 1.00 (1.00 to 1.00) | | |
| | 2 | | 1.00 (1.00 to 1.01) | | |
| | 3–4 | | 1.00 (0.99 to 1.01) | | |
| | 5–6 | | 1.00 (0.99 to 1.02) | | |
| | 7–10 | | 1.01 (0.98 to 1.03) | | |
| | 11–20 | | 1.01 (0.97 to 1.05) | | |
| | >20 | | 1.31 (0.48 to 3.52) | | |
| Age at first breakthrough seizure (linear) | ≤20 | 0.39 | 1.00 | N/A | N/A |
| | 21–30 | | 1.02 (0.97 to 1.07) | | |
| | 31–45 | | 1.07 (0.92 to 1.23) | | |
| | 46–70 | | 1.14 (0.85 to 1.53) | | |
| | >70 | | 1.22 (0.78 to 1.89) | | |

Continued

**Table 3** Continued

| Variable | Comparison | Univariable p value | Univariable HR (95% CI) | Multivariable HR (95% CI) | Multivariable HR (95% CI) w/o decision variable |
|---|---|---|---|---|---|
| Time to achieve initial 12-month remission (years) (FP) | 1 | <0.001 | 1.00 | 1.00 | 1.00 |
| | 1–1.5 | | 1.27 (1.12 to 1.44) | 1.21 (1.06 to 1.38) | 1.24 (1.08 to 1.41) |
| | 1.5–2 | | 1.57 (1.24 to 1.98) | 1.43 (1.12 to 1.82) | 1.49 (1.16 to 1.89) |
| | 2–3 | | 1.75 (1.31 to 2.34) | 1.56 (1.15 to 2.11) | 1.64 (1.21 to 2.22) |
| | >3 | | 1.89 (1.36 to 2.62) | 1.65 (1.17 to 2.43) | 1.75 (1.24 to 2.46) |
| Breakthrough seizure decision | No change to treatment plan | | 1.00 | 1.00 | N/A |
| | Increase dosage | <0.001 | 2.05 (1.59 to 2.66) | 2.05 (1.59 to 2.66) | |
| | Decrease dosage (or not specified) | 0.83 | 1.07 (0.59 to 1.93) | 0.99 (0.55 to 1.79) | |

Gen, generalised; HR >1, seizure recurrence more likely; FP, fractional polynomial transformation of this covariate; linear, no transformation of this covariate; N/A, variable not included in final model; TC, tonic–clonic.

quickly, and those patients who require only one drug to achieve remission, had a lower chance of a seizure recurrence.

The decision to not change antiepileptic drug dose following a breakthrough seizure was associated with a lower risk of a recurrence than the decision to increase dosage. This result is potentially counterintuitive as one might expect an increase in dose to reduce seizure risk. However, it is likely that clinicians are able to identify patients at higher risk of recurrence and recommend treatment changes to reduce that risk, although additional relevant clinical factors have not been identified by our model, and this requires further investigation. It is important to highlight that in most cases the decision to increase dose was taken in between neurology clinic appointments at which follow-up data were collected, presumably at the advice of the general practitioner or neurologist. As a result, accurate dates of dose increase have not been recorded and it is possible that a subgroup of patients had further seizures following the initial breakthrough seizure, prompting the clinician to increase the antiepileptic drug dose. When breakthrough seizure treatment decision was removed from the list of candidate variables to reflect the fact that clinicians will find it very difficult to use this information to inform treatment decisions for future patients, the parsimonious model included covariates for number of drugs required to achieve an initial 12-month remission and time taken to achieve initial 12-month remission. Only patients requiring polytherapy to achieve initial 12-month remission and taking at least 2 years to achieve initial 12-month remission required >12 months for their risk of a subsequent seizure to be <20%. This suggests that the current 12-month time off driving is generally appropriate. Even in the high-risk groups, the recurrence risks are fairly close to 20% if the focus is on point estimates.

Few publications have considered risk of a breakthrough seizure and tend to be focused on patients in the low-income, middle-income countries.[29 30] A study of 256 patients in Uganda identified non-compliance to antiepileptic drug therapy, duration of treatment, infections and menses among female study participants as factors significantly associated with breakthrough seizures.[30] Precipitating factors for breakthrough seizures for a study of 90 patients in Egypt were missed doses, sleep deprivation and psychological stress, although the authors also found differences in duration of seizure control, number of antiepileptic drugs and abnormal epileptic activity in EEG between patients with and without breakthrough seizures.[29] These factors were not collected as part of the SANAD study and as such have not been considered as part of this analysis. Neither study considered outcomes following the breakthrough study. We are unaware of any studies looking at outcome after a breakthrough seizure. In particular, we have been unable to identify any prognostic models considering risk of seizure recurrence following a breakthrough seizure for patients of driving age in developed countries. Another analysis of SANAD for patients of driving age has considered risk of a second treatment failure after a first.[31]

Others who have investigated driving regulations for patients with epilepsy have considered the time off driving required until the risk of seizure recurrence falls <2.5% per month.[32] This corresponds to a monthly risk of a seizure while driving of 1.04 per thousand and equates to 8 months off driving following an unprovoked first-ever seizure. Although the outcome under consideration in this article is breakthrough seizure after remission rather than first-ever seizure, the time off driving is fairly consistent across the papers.

## Limitations

SANAD recruited a large number of patients and followed them up for a long period—up to 6 years in some cases. However, only a small subset of these patients was relevant to address the question of risk of a seizure recurrence following a breakthrough seizure for patients of driving age. The requirement of patients to achieve

**Table 4** Risk of seizure recurrence in next 12 months estimated from multivariable model at specific seizure-free periods

| Patient characteristics | | | | Months of seizure freedom required from breakthrough seizure until annual risk falls<20% |
| Drugs required to achieve remission prior to breakthrough seizure | Time to achieve 12-month remission (years) prior to breakthrough seizure | Duration of seizure freedom after breakthrough seizure (months) | Risk of seizure in next 12 months (%, 95% CI) | |
| --- | --- | --- | --- | --- |
| Monotherapy | 1 | 6 | 20 (10 to 31) | 6.1 |
| | | 9 | 15 (4 to 25) | |
| | | 12 | 10 (0 to 21) | |
| | | 18 | 6 (0 to 16) | |
| | 2 | 6 | 30 (21 to 39) | 10.6 |
| | | 9 | 22 (13 to 32) | |
| | | 12 | 16 (6 to 26) | |
| | | 18 | 10 (0 to 19) | |
| | 3 | 6 | 32 (23 to 41) | 11.1 |
| | | 9 | 24 (15 to 33) | |
| | | 12 | 17 (8 to 27) | |
| | | 18 | 11 (1 to 20) | |
| | 4 | 6 | 33 (24 to 42) | 11.1 |
| | | 9 | 25 (16 to 34) | |
| | | 12 | 18 (8 to 27) | |
| | | 18 | 11 (1 to 20) | |
| Polytherapy | 2 | 6 | 37 (29 to 45) | 13.2 |
| | | 9 | 28 (19 to 30) | |
| | | 12 | 20 (11 to 30) | |
| | | 18 | 12 (3 to 22) | |
| | 3 | 6 | 40 (32 to 48) | 15.0 |
| | | 9 | 30 (22 to 39) | |
| | | 12 | 22 (13 to 31) | |
| | | 18 | 13 (4 to 23) | |
| | 4 | 6 | 41 (33 to 48) | 15.8 |
| | | 9 | 31 (22 to 39) | |
| | | 12 | 22 (13 to 31) | |
| | | 18 | 14 (5 to 23) | |

initial remission of at least 12 months and then have a breakthrough seizure to be included in this analysis also meant that the follow-up of patients after the breakthrough seizure was relatively short. This means that some CIs associated with the risk estimates are quite wide. Additionally, the SANAD data largely reflect patients with newly diagnosed epilepsy. We have therefore been unable to explore longer-term patterns. For example, if patients go into and out of remission then their seizure recurrence risks might change compared with these estimates. The subset of patients considered for this analysis may also have limited power to detect some prognostic effects as significant. Other important factors may exist which have not been analysed or collected. The SANAD study also indicated that lamotrigine was superior to carbamazepine in terms of seizure control for partial onset seizures.[10] Given the relatively small sample size, we have had to combine treatment groups for our analysis rather than undertake per-drug analyses and thus assume that combining groups is clinically valid.

The multivariable model for risk of seizure recurrence included a continuous covariate—time to achieve initial 12-month remission. Therefore, to estimate the risk of recurrence over the next 12 months for combinations of risk factors including this covariate, the variable had to be categorised which may not be the most efficient approach.[33] Also, neurological insult, seizure type, epilepsy type and CT/MRI scan result were recorded at baseline rather than at the breakthrough seizure. Although these covariates may have changed by a breakthrough seizure, it is likely that any change occurred in only a small number of patients. EEG was also only recorded at baseline, and

it is possible that EEG on treatment would be prognostic, although given the unpredictable nature of breakthrough seizures, it would not be feasible to undertake an EEG in order to inform risk.

There is evidence to suggest that patients with epilepsy may elect not to report breakthrough seizures to their clinicians or the relevant driving authority.[34] The evidence collected as part of SANAD is patient-reported seizure counts and therefore our results may be under-estimating the actual risk. Increased patient counselling regarding the risks involved with driving, the need for driving regulations and the importance of compliance with these rules may only have a limited impact as the implications for patients losing their driving licence are potentially serious such as job losses and resulting lack of independence. The model developed here should ideally be validated in other similar data sets. However, no other similar data sets exist. The best match is a set of individual participant data we have collected.[35] These data include only very small numbers of relevant patients. Therefore, alternative data sources are required.

## CONCLUSIONS

Twelve months appears to be an appropriate time off driving for patients of driving age who have experienced a period of at least 12 months initial seizure freedom followed by a breakthrough seizure. Provided that patients remain seizure free for 12 months following a breakthrough seizure, their risk of a seizure in the next 12 months would be less than the 20% risk standard that informs the UK legislation and DLVA guidance.

As discussed in depth in Bonnett,[7] the legislators and DVLA need to decide whether to base time off driving on unadjusted estimates only or whether they should consider estimates adjusted for important clinical factors. Although our unadjusted results suggest that 12 months off driving is sufficient time off driving, risk estimates differ substantially among groups. For some patient subgroups, at least 15 months off driving is required for their point estimate to reduce <20%. Additionally, discussions are required to determine whether associated 95% CIs should be used to inform the decision-making process. The unadjusted risk estimate is significantly <20% by 12 months. However, none of the adjusted risk estimates are significantly <20% by 12 months.

Evidence is inconclusive regarding whether drivers with epilepsy have higher rates of motor vehicle accidents than those without epilepsy. However, there is evidence that accidents are 26 times more likely to occur with drivers with other medical conditions compared with drivers with epilepsy.[36] Implementing a policy based on clinical factors is potentially challenging. In fact, in practice time to achieve remission may be the only factor that could be incorporated into such an assessment as there is potential for manipulation of drugs in terms of number and doses to meet driving objectives. Furthermore, introducing a tiered system may compromise patient care as patients would be inclined to 'fit in' to the shorter duration if driving is important to them.

**Contributors** LJB undertook all analyses presented in this manuscript. AGM, CTS and LJB developed the analysis plan and interpreted the analysis results. GP extracted required additional information from the SANAD patient case report forms. All authors drafted and redrafted the manuscript. AGM is the guarantor for this work.

**Funding** This report is independent research arising from a postdoctoral fellowship (LJB, PDF-2015-08-044) supported by the National Institute for Health Research. AGM is part funded by National Institute for Health Research Collaboration for Leadership in Applied Health Research and Care North West Coast (NIHR CLAHRC NWC).

**Disclaimer** The views expressed in this publication are those of the authors and not necessarily those of the NHS, the National Institute for Health Research or the Department of Health.

**Competing interests** None declared.

**Provenance and peer review** Not commissioned; externally peer reviewed.

**Data sharing statement** The anonymised individual participant data from the SANAD study will be made available for research purposes by contacting AGM. Statistical code is available on request from the corresponding author.

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
