## [Reviewer comments · BMJ Open]

ARTICLE DETAILS

TITLE (PROVISIONAL)	Risk of a seizure recurrence after a breakthrough seizure and the implications for driving - further analysis of the Standard versus New Antiepileptic Drugs (SANAD) Randomised Controlled Trial
AUTHORS	Bonnett, L; Powell, Graham; Tudur-Smith, Catrin; Marson, Anthony

VERSION 1 - REVIEW

REVIEWER	Joon Kang Johns Hopkins School of Medicine United States
REVIEW RETURNED	26-Jan-2017

GENERAL COMMENTS	In this manuscript, Dr. Bonnett and colleagues analyze the data from the Standard and New Antiepileptic Drugs (SANAD) to provide data regarding seizure recurrence risk after a breakthrough seizure (defined as at least 12 months initial seizure freedom). The authors used regression modeling to investigate whether a number of clinical factors and number of antiepileptic drugs influenced seizure recurrence risk. Overall, the paper was well written and addressed the research questions. The limitations are that a small fraction of total patients in the SANAD trial had breakthrough seizure (figure 1), the follow up was relatively short (median of 1.67 years and median 1.41 years for arm A and arm B) and the trial largely included new onset epilepsy patients. The findings from this study are probably not accurate in patients who have chronic epilepsy with occasional breakthrough seizures, but the authors acknowledge this limitation. In summary, this is a clinical useful paper that highlights the need for more evidence based driving mandates.
--

REVIEWER	David Carr Washington University at St. Louis USA
REVIEW RETURNED	01-Feb-2017

GENERAL COMMENTS	A very well done study and quite informative in an area we need more research and data to provide guidelines in the area of driving and epilepsy. Intro: Would spell out SANAD for the first time in the text even if done in the abstract Might consider adding other countries guidelines or the range required to be seizure free as a last sentence to the first paragraph. Would be good to spell out what the risks were in the Bonnett
---

	studies. Methods: The abstract discusses the Arm's and implies they were developed based on types of seizures while the Methodology based the Arm's in the SANAD based on the clinicians choice of primary drug. Would be consistent. The authors make the assumption that combining groups make sense, yet the SANAD study indicated there was a difference in AED outcomes (lamotrigine superior to carbamazepine) at least for partial onset seizures. Is the overall N too small to do an analysis based on AED used? I suspect so, but this should be mentioned as a limitation in the Discussion section. Results: Table 2, third column, is this the N, % or RR? please define better. I like the information in Table 3, but if space an issue, discussing pertinent aspects in text form would be fine. The paragraph "Breakthrough seizure treatment decision,..." would be better off left to the Discussion section. Discussion. The authors have an excellent discussion section and Limitations section and I have little to add. I like how they make the case for the 12 month wait post breakthrough... even though it may take a little longer to reduce risk below 20% for those on polytherapy and needing 3-4 years to achieve 12 month remission. You might add to the "safety" discussion by pointing out the low crash risk of those with epilepsy compared to controls, especially with newer reviews (Nalk et al. Epilepsy and Behavior, 2015. It might also be interesting to compare your data and your outcome (<20% risk) to the Brown JW et al Journal of Neurology, Neurosurgery, and Psychiatry 2015 study...although they focused on first ever seizures...they used the bar of a rate falling <2.5% per month which is pretty consistent with this paper.
--	--

REVIEWER	Chiara Di Gravio MRC Lifecourse Epidemiology Unit University of Southampton United Kingdom
REVIEW RETURNED	19-Feb-2017

GENERAL COMMENTS	The statistical approach is appropriate given the question posed by the authors. The methodology used is explained well and in detail. Below are some minor comments/suggestions for the authors. Minor Comment:  - Might be worth adding the breakdown of seizure recurrence after breakthrough seizure (e.g. what is the percentage of people who had an event after 1 month, 2 month, 6 month, one year and so on.) - The consort diagram in Figure 1 gives the impression that the two arms were analysed separately.
--

REVIEWER	Deborah Ridout UCL Institute of Child Health
REVIEW RETURNED	28-Feb-2017

GENERAL COMMENTS	This is a very interesting and well conducted study, for which the background has been introduced and described well. I have a few
--

	comments:  1. The statistical analysis section is quite difficult to follow and I wonder if it could be structured differently, maybe practical data issues and the technical aspects could be separated more. Furthermore it is not clear where log transformations and fractional polynomials were used and similarly where a spline model was fitted. This could be described briefly in the results section. 2. Model estimates are provided in table 4 in a succinct format, but it may be difficult for the reader to interpret these results – particularly column 3. 3. Is it possible to provide confidence intervals for the estimates in column 5 of table 4? 4. We are given no indication how well the model fits the data
--	--

VERSION 1 – AUTHOR RESPONSE

Reviewer 1

1 No comments requiring action

We thank the reviewer for their kind comments and recognise the limitations they raise, which we have acknowledged within the manuscript, as confirmed by this reviewer.

Reviewer 2

1 Would spell out SANAD for the first time in the text even if done in the abstract.

We have now added the full study name next to the first use of SANAD within the Introduction.

2 Might consider adding other countries guidelines or the range required to be seizure free as a last sentence to the first paragraph.

We have added three brief sentences to the end of the first paragraph of the Introduction outlining the guidelines within other European countries, and also within the United States.

3 Would be good to spell out what the risks were in the Bonnett studies.

We have briefly summarised the risks calculated as part of the two Bonnett studies within the Introduction section of the manuscript.

4 The abstract discusses the Arm's and implies they were developed based on types of seizures while the Methodology says the Arm's in the SANAD were based on the clinicians choice of primary drug. Would be consistent.

We thank the reviewer for noticing this inconsistency, although we believe it is between the Methodology and Introduction sections rather than the Abstract. The description within the Methodology is the correct description of the process. We have therefore modified the Introduction to better reflect this.

5 The authors make the assumption that combining groups makes sense, yet the SANAD study indicated there was a difference in AED outcomes (lamotrigine superior to carbamazepine) at least for partial onset seizures. Is the overall N too small to do an analysis based on AED used? I suspect so, but this should be mentioned as a limitation in the Discussion section.

The reviewer is correct in their suspicion that the sample size is too small to consider each drug independently. We have therefore added the suggested limitation to the Discussion section.

6 Table 2, third column, is this the N, % or RR? Please define better.

We have added the units (%) to column three of Table 2 to help define these values better.

7 I like the information in Table 3, but if space is an issue, discussing pertinent aspects in text form would be fine.

The Editor has not commented on space. Therefore, we have left Table 3 unaltered as we feel it clearly outlines risks based on patient subgroups.

8 The paragraph "Breakthrough seizure treatment decision,..." would be better off left to the Discussion section.

We do understand the reviewer's logic for this suggestion. However, given that Table 3 includes two columns – one for with treatment decision and one for without treatment decision we feel that it is more appropriate in its current location. However, we have added a link to Table 3 which was previously absent which we feel better justifies the inclusion of the text in question in its current location.

9 You might add to the "safety" discussion by pointing out the low crash risk of those with epilepsy compared to controls, especially with newer reviews (Nalk et al).

We thank the reviewer for making us aware of this reference. We have added this to the Discussion section.

10 It might also be interesting to compare your data and your outcome (<20% risk) to the Drown JW et al study...although they focused on first ever seizures...they used the bar of a rate falling <2.5% per month which is pretty consistent with this paper.

Again we thank the reviewer for introducing us to this recent publication. Despite the differences in the focus of the two manuscripts we have included the recommended reference and compared their findings with ours.

Reviewer 3

1 Might be worth adding the breakdown of seizure recurrence after breakthrough seizure (e.g. what is the percentage of people who had an event after 1 month, 2 month, 6 month, one year and so on).

We have now added such a breakdown to the Results section.

2 The consort diagram in Figure 1 gives the impression that the two arms were analysed separately.

To help avoid this confusion we have reminded readers that data from both arms have been combined for this analysis. This is now within the Results section when we mention Figure 1. We feel it is appropriate to keep Figure 1 as it is because the original SANAD trial was conceived for both arms to be analysed separately. We have discussed the limitations of the pooling we have undertaken in the Discussions section.

Reviewer 4

1 The statistical analysis section is quite difficult to follow and I wonder if it could be structured differently, maybe practical data issues and the technical aspects could be separated more.

We have re-ordered the Statistical Analysis section so that the description of methods is presented together, followed by a description of the data handling methodology.

2 Furthermore it is not clear where log transformations and fractional polynomials were used and similarly where a spline model was fitted. This could be described briefly in the results section.

As described in the Statistical Analysis section, the spline model was only fitted to the continuous covariates following the analyses. This was done solely to determine suitable cut points (knots) for presentation of the continuous covariates as categorical variables within the Results tables.

We have added a footnote to Table 3 which describes which transformation was used for each of the

continuous covariates.

3 Model estimates are provided in table 4 in a succinct format, but it may be difficult for the reader to interpret these results – particularly column 3.

We appreciate the challenges of interpreting Table 4. Therefore, we have modified the column titles to assist with this.

4 Is it possible to provide confidence intervals for the estimates in column 5 of table 4?

Unfortunately we are unable to provide confidence intervals for the estimated duration until annual risk falls below 20%. These estimates have been obtained by calculating the annual recurrence risk at each event time, and observing the first time point at which all subsequent risks are less than 20%.

5 We are given no indication how well the model fits the data.

We apologise for this oversight. We have now assessed the predictive accuracy of the model using the c-statistic and have thus updated both the Methods and Results sections accordingly.

We hope that these changes to the manuscript will be to the reviewers' satisfaction.

VERSION 2 – REVIEW

REVIEWER	Deborah Ridout UCL Institute of Child Health
REVIEW RETURNED	12-Apr-2017

GENERAL COMMENTS	The authors have address all my earlier queries and I have no further queries
---